# A Critical, Temporal Analysis of Saudi Arabia's Initiatives for Greenhouse Gas Emissions Reduction in the Energy Sector

**Muhammad Muhitur Rahman** [1,*], **Md Arif Hasan** [2], **Md Shafiullah** [3,*], **Mohammad Shahedur Rahman** [4], **Md Arifuzzaman** [1], **Md. Kamrul Islam** [1], **Mohammed Monirul Islam** [5] **and Syed Masiur Rahman** [6]

1   Department of Civil and Environmental Engineering, College of Engineering, King Faisal University, Al-Ahsa 31982, Saudi Arabia
2   Climate Change Response Unit, Wellington City Council, 113 The Terrace, Wellington 6011, New Zealand
3   Interdisciplinary Research Center for Renewable Energy and Power Systems, King Fahd University of Petroleum & Minerals, Dhahran 31261, Saudi Arabia
4   Civil Engineering Department, College of Engineering, Imam Mohammad Ibn Saud Islamic University, Riyadh 13318, Saudi Arabia
5   Department of Biomedical Sciences, College of Clinical Pharmacy, King Faisal University, Al-Ahsa 31982, Saudi Arabia
6   Applied Research Center for Environment & Marine Studies, King Fahd University of Petroleum & Minerals, Dhahran 31261, Saudi Arabia
*   Correspondence: mrahman@kfu.edu.sa (M.M.R.); shafiullah@kfupm.edu.sa (M.S.)

**Abstract:** The per capita greenhouse gas (GHG) emissions of Saudi Arabia were more than three times the global average emissions in 2019. The energy sector is the most dominant GHG-emitting sector in the country; its energy consumption has increased over five times in the last four decades, from over 2000 quadrillion joules in 1981 to around 11,000 quadrillion joules in 2019, while the share of renewable energy in 2019 was only 0.1%. To reduce GHG emissions, the Saudi Arabian government has undertaken initiatives for improving energy efficiency and increasing the production of renewable energies in the country. However, there are few investigative studies into the effectiveness of these initiatives in improving energy efficiency and reducing greenhouse gas emissions. This study provides an overview of the various energy efficiency and renewable energy initiatives undertaken in Saudi Arabia. Then, it evaluates the effectiveness of energy-related policies and initiatives using an indicator-based approach. In addition, this study performs temporal and econometrics analyses to understand the trends and the causal relationships among various drivers of energy sector emissions. Energy intensity and efficiency have improved moderately in recent years. This study will support policymakers in identifying significant policy gaps in reducing the emissions from the energy sector; furthermore, this study will provide a reference for tracking the progress of their policy initiatives. In addition, the methodology used in this study could be applied in other studies to evaluate various climate change policies and their progress.

**Keywords:** decarbonization; demand management; econometric analysis; energy sector; energy efficiency; GCC country; greenhouse gas emissions; renewable energy; Saudi Arabia

## 1. Introduction

According to the World Bank [1], global per capita $CO_2$ emission values in 2019 were 4.5 tonnes; meanwhile, Saudi Arabia's per capita $CO_2$ emission values at that time were 15.3 tonnes (i.e., more than three times the global average). However, The Kingdom of Saudi Arabia has recently demonstrated its commitment to reach net-zero emissions by 2060 and has announced an investment of over USD 180 billion to achieve this emission-reduction target [2]. It also joined the Global Methane Initiative (GMI) and committed to reducing global methane emissions by 30% between 2020 and 2030. In addition, The Kingdom has committed to increasing the contribution of renewable energies to its electrical mix by

50 percent before 2030 and improving energy efficiency [3]. However, The Kingdom's remarkable economic and industrial development has resulted in a massive increase in domestic energy usage over the past several decades. Based on current patterns, domestic energy consumption in the country is expected to increase by between 4% and 5% per year through 2030. Such a surge in demand indicates the economic prosperity and industrial progress of The Kingdom of Saudi Arabia. Figure 1 shows Saudi Arabia's total energy consumption over the past four decades from different sources.

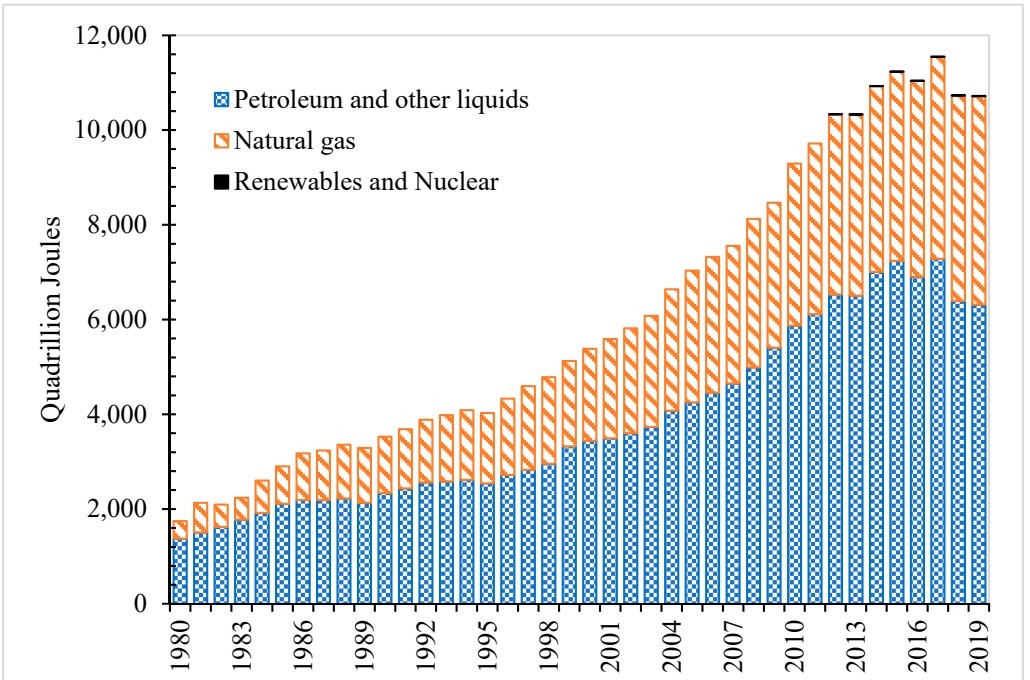

**Figure 1.** Total energy consumption from various energy sources (adapted from [4]).

Due to the rapid expansion in The Kingdom's energy demand, energy efficiency has become a primary consideration in all key choices. The Kingdom has recently undertaken numerous programs to promote energy productivity and decrease energy usage. The National Energy Efficiency Program (NEEP) aims to improve the management and efficiency of power generation and consumption [5]. Under the NEEP, the Saudi Energy Efficiency Center (SEEC) was founded. The primary aims of the SEEC were to (i) formulate a national energy efficiency program, (ii) propose and monitor energy efficiency rules and regulations, (iii) raise awareness about energy efficiency, and (iv) participate in pilot project implementation. The SEEC also started the Saudi Energy Efficiency Program (SEEP) in 2012, which engages with government agencies that make choices about urban planning and focuses on product control mechanisms and enforcement to assure compliance with new regulations and standards for the energy, building, and transport sectors [6]. In addition, public awareness initiatives educate the public on the rationale for regulatory and standards changes. With the cooperation of the Ministry of Finance, SEEP devised a systematic mechanism for selecting financial-incentive-based initiatives. Additionally, it reached an arrangement with the Saudi Industrial Development Fund to provide soft loans for energy-efficiency-related industrial projects. The initiative has been developing techniques to facilitate the formation of National Energy Services Companies (NESCO), which includes an accreditation system, a measurement and verification process, and a standard for energy services performance contracts [6].

Apart from the various energy efficiency initiatives, The Kingdom has also undertaken various renewable energy initiatives to increase the share of renewable energy in the energy mix from less than 1% to 30% by 2030. In 2016, the King Salman Renewable Energy Initiative was unveiled, which intends to (i) evaluate the regulatory system that permits

the private sector to purchase and invest in renewable energy, (ii) localize the sector and create the right skill sets through public–private partnerships, and (iii) ensure renewable energy's competitiveness through gradual market liberalization [7]. In addition, the country established the National Renewable Energy Program (NREP), a long-term strategic project being implemented in tandem with Saudi Vision 2030 and the King Salman Renewable Energy Initiative. The Kingdom is also trying to broaden its energy mix and is actively considering nuclear energy projects. As part of this initiative, the King Abdullah City for Atomic and Renewable Energy (KACARE) established plans for the atomic energy industry and announced the Saudi National Atomic Energy Project in 2017 [8]. The project's primary benefits include (i) increasing energy source diversification, (ii) utilizing atomic energy for saltwater desalination, and (iii) creating new learning, training, and job opportunities in the field of renewable and alternative energy.

In order to achieve the targets of the renewable energy and energy efficiency initiatives, the country has been investing a significant amount of money in research and development (R&D)—over SAR (Saudi Arabian Riyal) 6 billion annually, equivalent to USD 1.6 billion [9]. In addition, The Kingdom has established several research centers at various universities to facilitate education, research, and development (Table 1). For instance, the King Abdullah University of Science and Technology (KAUST) Solar Center seeks to advance science and technology in solar energy conversion by fostering interdisciplinary research, education, and innovation for the benefit of society. Additionally, the King Fahd University of Petroleum, the Minerals' Center of Research Excellence in Renewable Energy, and the King Abdulaziz University's Center of Research Excellence in Renewable Energy and Power Systems have been collaborating to advance current and emerging sustainable, renewable energy technologies through distinguished research in support of Saudi Arabia's Vision 2030. Moreover, several public and private organizations, including Saudi Aramco and Saudi Electric Company (SEC), have been working in the energy efficiency and renewable energy sectors [9].

**Table 1.** Energy efficiency and renewable energy research centers in Saudi Arabia (adapted from [9]).

| Institution | Centers |
| --- | --- |
| King Abdullah University of Science and Technology (KAUST) | KAUST Solar Center<br>Energy Conversion Devices and Materials (ECO Devices) Laboratory |
| King Fahd University of Petroleum and Minerals (KFUPM)<br>King Saud University (KSU)<br>King Abdulaziz University (KAU)<br>Saudi Electricity Company (SEC)<br>Mishkat Interactive Center | Center of Research Excellence in Renewable Energy<br>Sustainable Energy Technologies Center<br>Center of Research Excellence in Renewable Energy and Power Systems<br>Research Center of Renewable and Energy Storage<br>Center for Atomic and Renewable Energy |

The above examples indicate that The Kingdom has undertaken various initiatives over the last decade to improve energy efficiency and increase the share of renewable energy in the energy mix. Several studies on Saudi Arabia's energy sector emissions have also highlighted the economic and environmental benefits of various energy efficiency and renewable energy initiatives [10–13]. However, none of these studies have evaluated the progress of these initiatives towards reducing the emissions of the energy sector. In addition, no study used tailor-made indicators to understand how effective these initiatives are in improving energy efficiency or enhancing the share of renewable energy for lowering GHG emissions.

Given that the challenges and policy gaps need to be investigated to ensure that The Kingdom is progressing well towards achieving its renewable energy, energy efficiency, and carbon reduction targets, this study explores the effectiveness of various energy efficiency and renewable energy initiatives and identifies the major policy gaps. The major contributions of the present article are as follows:

- It provides an overview of the various energy efficiency and renewable energy initiatives taken by The Kingdom of Saudi Arabia for reduction of GHG emissions.



- It evaluates the effectiveness of the discussed initiatives and policies using an indicator-based approach. In addition, this study performs temporal and econometrics analyses to understand the trends and the causal relationships among various drivers of energy sector emissions.
- The findings of the study will support policymakers in identifying significant policy gaps in reducing emissions from the energy sector and tracking the progress of their policy initiatives.

The following section explores various energy efficiency and renewable energy initiatives undertaken by The Kingdom, while Section 3 provides a methodology to investigate the effectiveness of these initiatives. Section 4 presents and discusses the results, and Section 5 identifies policy gaps and implications. Section 6 presents the conclusions of this study.

## 2. Energy Efficiency and Renewable Energy Initiatives

This section analyzes the initiatives taken by The Kingdom to improve energy efficiency and increase the share of renewable energy in the energy mix.

### 2.1. Energy Efficiency Initiatives

The government is currently collaborating with key stakeholders and consolidating efforts to develop a mechanism of action to increase the energy efficiency of the utility sectors in The Kingdom of Saudi Arabia, which include: (i) electricity generation and co-generation plants, (ii) desalination plants, and (iii) electricity transmission and distribution networks. Some of the works carried out under this initiative include (i) identifying relevant stakeholders; (ii) engaging relevant sector entities such as service providers and technology providers; (iii) developing the key performance indicators (KPIs); (iv) establishing the sector's baseline energy efficiency; (v) comparing the sector's baseline energy efficiency to that of other sectors; and (vi) the construction of an action mechanism to increase energy efficiency [14].

To improve energy efficiency, the government has invested in modernizing and upgrading the entire power sector, including replacing aging power plants, infrastructure upgrades, implementing smart grid technology, installing smart meters, and promoting international grid connectivity [15]. These initiatives are presented in Table 2, and the effectiveness of these initiatives is analyzed using the energy intensity indicator (i.e., energy consumption per unit of GDP) outlined in Section 3.

In addition, The Kingdom has been working to build a competitive energy market by establishing an independent transmission company that will operate on an open and unbiased basis for all producers and large customers [16]. Apart from establishing the national electricity market, The Kingdom recently led a GCC effort to connect the member nations' power grids to facilitate electricity trading and better manage peak load demand. The Kingdom has already begun constructing an HVDC link with Egypt to share three gigawatts (GW) of electricity. Additionally, The Kingdom is interested in connecting its power grid to Turkey, Yemen, and other Arab states to facilitate electricity trading, reduce peak loads, and increase system efficiency [15].

In addition, the Saudi Electricity Company (SEC) raised combined-cycle output from 8.3% in 2010 to 31.0% in 2019, whereas simple-cycle production declined from 50% in 2010 to 22% in 2019 [17]. The combined cycle generated a total of 59,258 MWh of energy. The business is actively implementing projects to increase the Al-Qassim, Duba, Hail, Waad Al-Shamal, Taibah, and Northern Al-Qassim plants by more than 4.2 gigawatts. It should be emphasized that these projects would result in around 60 million barrels of oil being saved annually and an approximate 25.8-million-ton reduction in carbon dioxide emissions. Additionally, the Rabigh 2 production project is a remarkable achievement for The Kingdom's power industry since it is the first independent project to use combined-cycle technology with an overall thermal efficiency of 58.8%. The project adds a total

capacity of 2060 MW to the grid [17]. As a result, the Saudi Arabian electricity distribution networks would be estimated to account for most of their renewable energy goals.

Inefficient electrical appliances and a lack of thermal insulation are key drivers of building energy consumption [18]. A series of projects are being undertaken to achieve efficiency in this sector, including a change in the standard specifications for air conditioners, lighting, and additional household equipment. In addition, The Kingdom supports the domestic production of high-efficiency air conditioners and offers incentives to stimulate sales [19]. The Kingdom has also developed many guidelines to rationalize energy consumption in buildings focusing on thermal insulation, domestic washing machines, refrigerators and freezers, air conditioning, lighting, and heaters [18]. A detailed list of various energy demand management initiatives is outlined in Table 3, and the effectiveness of these initiatives is measured through analyzing the per capita energy consumption trend.

The petrochemical, cement, and steel industries consume around 70% of the total industrial energy used, so energy efficiency programs are focusing on these key industries [6]. The Saudi Energy Efficiency Center (SEEC) established energy efficiency standards and criteria for the iron, cement, and petrochemical industries and built an Energy Reporting System website to collect pertinent data annually from businesses and energy vendors. The SEEC is currently working to issue annual reports for the listed manufacturers; these will outline their yearly performance and progress, and identify any potential concerns. In addition, the Saudi Electricity Company has recently started charging industrial, commercial, and governmental customers for inefficient energy consumption rates. It is expected that the introduction of fees for inefficient energy consumption rates will bring efficiency to every major sector, including the petrochemical, cement, and steel industries.

## 2.2. Renewable and Nuclear Energy Initiatives

The Kingdom has committed to a 30% increase in the share of renewable energy in the national energy mix by 2030 [20]. Saudi Vision 2030 calls for the establishment of a competitive renewable energy sector. The NREP is a long-term strategic project implemented with Vision 2030 and the King Salman Renewable Energy Initiative. The program attempts to maximize Saudi Arabia's renewable energy potential. The Ministry of Energy oversees this program, which began in 2017. The program aims to build and advance the renewable energy industry by leveraging private sector investment and encouraging public–private collaborations. The NREP will diversify Saudi Arabia's energy mix, reducing The Kingdom's reliance on oil and GHG emissions following the Paris Agreement; it will also enable job creation and drive economic development throughout The Kingdom, ensuring long-term prosperity.

To meet expanding electricity demand, sustain economic growth, and diversify the domestic energy mix, The Kingdom has adopted sophisticated and cost-effective technologies to satisfy the high demand, including creating 46 solar energy measurement stations throughout The Kingdom. The daily average of direct normal irradiance (DNI), which is used to generate concentrated solar power (CSP), and the daily average of diffused horizontal irradiance (DHI), which is used to generate diffused horizontal irradiance (DHI), are around 5000 Wh/m$^2$/day and 2000 Wh/m$^2$/day, respectively. All the indicators (such as GHI, DNI, and DHI) demonstrate the enormous potential of solar energy technologies (both photovoltaic and concentrated solar power) throughout The Kingdom [21]. Additionally, the country has erected ten stations throughout The Kingdom to assess the viability of wind energy systems. At the height of 40 m, the average annual wind speed was reported to be greater than 5 m/s. The country has even higher wind speeds at 60-m, 80-m, 98-m, and 100-m altitudes. Wind speed observations from a few selected places show a significant potential for wind energy systems in several locations throughout The Kingdom [21]. Table 2 presents the selected locations of the various renewable energy projects in The Kingdom.

By the end of 2018, The Kingdom had deployed 89 MW of solar PV, 50 MW of CSP, and 3 MW of wind energy [22]. The Dubai Independent Power Producer (IPP) was awarded a contract to construct an integrated solar combined cycle power plant named Green Duba,

with a projected capacity of over 605 MW, including a 50 MW integrated solar facility [23]. Saudi Arabia's first utility-scale solar photovoltaic (PV) facility, the Sakaka 300 MW solar power plant, was connected to the national grid by the end of 2019 [24]. The Public Investment Fund (PIF) is also expected to construct several renewable energy projects throughout The Kingdom in collaboration with a strategic partner [25,26]. Negotiations with the Public Investment Fund are underway to create the 2.0 GW Sudair Solar Project, authorized by Royal Decree No. 5629, dated 27/1/1441H [17]. The Kingdom has already selected the contractor for Sudair Project's 1.09 GW DC solar plant [27].

**Table 2.** Renewable energy projects in The Kingdom (adapted from [28,29]).

| Type | Locations |
|---|---|
| Solar PV | Al-Faisalia, Al-Haeer, Al-Kharj, Al-Laith, Al Masa'a, Al-Quwaiiyah, Bisha, Dhahban, Dhurma, Farasan, Ghilanah, Haden, Henakiyah, Jazan, Layla, Madinah, Mahd Al-Dhab, Malham, Mastoorah, Qaisumah, Qurayyat, Rabigh, Rafha, Sakaka, Sharorah, South Jeddah, South Yanbu, Sudair, Tabarjal, Tuwaiq, Unaizah, Wadi Al-Dawasir. |
| Wind | Al-Ghat, Al-Ras, Dumat Al-Jandal, Duwadimi, Shaqra, Sourah, Starah, Tuwaiq, Waad Al-Shammal, Wadi Al-Dawasir, Yanbu. |
| CSP | Al-Kahafa, Khushaybi, Tabarjal, Tabuk. |

Saudi Aramco is also investing significantly in various renewable energy initiatives. Between 2014 and 2016, Saudi Aramco developed technical, planning, and commercial capabilities in renewable energy, including analyzing over 380 project areas and surveying 75 domestic and 131 international manufacturers [30]. In January 2017, Saudi Aramco partnered with General Electric (GE) to install a 2.75-megawatt wind turbine at Turaif Bulk Plant. In addition, a 500-kilowatt solar farm was erected on Farasan Island in the Red Sea in partnership with the SEC and Solar Frontier, Aramco's equity partner with Showa Shell of Japan.

Like Saudi Aramco, the Saudi Electric Company (SEC) works on various renewable projects. For example, the SEC finished the first phase of the Layla Al-Aflaj solar energy project, which has a capacity of 10 MW and is directly connected to the utility grid. Additionally, the SEC has developed a 500 kW solar power facility on Farasan Island in Jazan, generating 860 GWh of electricity yearly. Annually, the plant saves around 565.1 tons of $CO_2$. Five solar-powered systems have been placed in several of the SEC's buildings to save roughly 8,550,000 kWh per year, or the equivalent of 15,400 barrels of oil [17]. Additionally, the SEC announced the launch of early applications for residential and commercial solar photovoltaic (PV) installations. [31]. In conjunction with King Saud University, the SEC inaugurated the world's first concentrated solar power (CSP) heliostat field in Riyadh Techno Valley.

The Kingdom is also considering scaling up its nuclear power facilities. The Kingdom established the King Abdullah City for Atomic and Renewable Energy (KACARE) to establish a sizable atomic and renewable energy capacity supported by local industry [32]. To safeguard the safety and security of nuclear power plants, an independent regulator is now being established inside KACARE, following international advice [8]. It is expected that nuclear power facilities will benefit The Kingdom by (i) increasing energy source diversification; (ii) extending the life of petroleum supplies; (iii) utilizing atomic energy for saltwater desalination; and (iii) creating new learning, training, and job opportunities.

Municipal solid waste (MSW), wastewater treatment plant (WWTP) byproducts, and industrial and agricultural organic waste have the potential to be harnessed for renewable energy in The Kingdom [33]. Although waste management is a persistent issue in Saudi Arabia, particularly in urban areas, waste-to-energy (WTE) projects could potentially reduce waste volumes while producing useful energy in a climate-friendly manner [34]. The country has enormous WTE potential due to its abundant supply of high-quality

MSW, and waste-to-energy conversion capacity should reach 3 GW by 2040 [35]. The Kingdom has established a USD 300 million waste-to-energy facility. The facility has processed approximately 180 tons of wastewater into distilled water and generates about 950 m$^3$ of water and 6 MW of electricity per day [35]. Additionally, as part of its National Transformation Program, The Kingdom has taken on the "Reuse Wastewater Initiative", which aims to increase the reuse of treated water [36]. Major industries, such as Saudi Aramco, are also taking numerous initiatives to manage industrial waste and treat water.

With its vast quantities of municipal garbage, agricultural residues, and agro-industrial waste, The Kingdom has the capacity to develop biomass energy technologies. Municipal solid waste is the primary source of biomass in the country, followed by gross urban waste. Food waste accounts for the third-largest component of waste generated. Additionally, the country generates massive amounts of sewage sludge daily. Furthermore, the country's food processing industry, animal population, and poultry sector have grown rapidly. All of these products are biomass energy sources [37]. However, the current management system involves simply collecting the waste and depositing it in an open landfill, creating environmental and public health problems [38]. Advanced biomass-conversion technologies can significantly reduce the environmental impact of a wide variety of biomass wastes by serving as a safe disposal method for solid and liquid biomass wastes and as an attractive option for generating heat, power, and fuels [37].

Saudi Arabia leads the world in desalinated water production and consumption, with 2558 million cubic meters produced in 2019, an increase from the 1129 million cubic meters produced in 2007 [39]. In addition, the country generates electricity using desalination plant water, one of the largest renewable energy sources. The total amount of electricity (i.e., hydroelectricity) generated by desalination plants for licensed companies such as SWCC reached a record high of approximately 151 thousand gigawatt-hours (TGWH) in 2017, up from about 99 TGWH in 2012 (Figure 2). However, the electricity generated in 2018 exhibited a decrease from the 2017 level by around 16 TGWH [21].

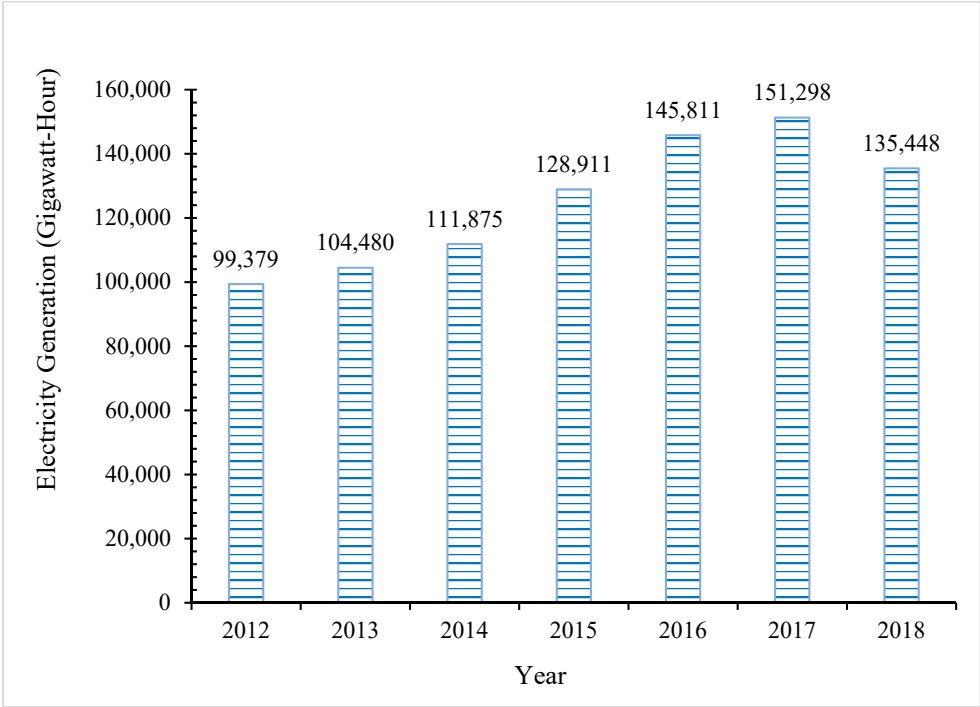

**Figure 2.** Total generated hydroelectricity in The Kingdom (adapted from [21]).

A summarized list of the renewable energy initiatives of The Kingdom of Saudi Arabia is presented in Table 3, and the effectiveness of these initiatives is examined using the emissions intensity indicator (i.e., greenhouse gas emissions per unit of energy consumption). The details of the methodologies used in this paper are presented in Section 3.

## 3. Methodology

This study evaluates the effectiveness of various energy-related policies in Saudi Arabia using an indicator-based approach. To understand the trend of each indicator, temporal analyses were conducted based on the double exponential smoothing method. In addition, this study performed an econometrics analysis to understand the causal relationships among various drivers of energy sector emissions. The details of the methodologies are discussed below.

### 3.1. Evaluating Energy Policies: Indicator-Based Approach

Researchers have used various methodologies to assess the effects of energy efficiency and renewable energy initiatives on emissions reduction. While some academics have evaluated energy policies primarily based on their sources, such as petrol, diesel, coal, and renewables [40–42], others have used an indicator-based approach [43,44]. This study used an indicator-based analytical approach (developed in Refs. [43,44]) to analyze The Kingdom's energy efficiency and renewable energy initiatives.

Policy initiatives related to energy efficiency and renewable energy were categorized under three broad policy outcomes: energy decarbonization, energy efficiency improvement, and energy demand management. Then, an indicator was chosen for each policy outcome to assess the effectiveness of the relevant energy-related policy initiatives. While choosing indicators, a few aspects were taken into consideration. Firstly, the data needed to allow the indicators to be made available in the required format. Secondly, the chosen indicators were made to be easily understandable and straightforward, so that other countries can apply them in tracking their own processes and detecting their own policy shortcomings. Thirdly, the indicators were made to be relevant to the policy outcomes. Finally, the indicators offer tailor-made options rather than energy sector emissions to clarify a country's policy portfolio and provide recommendations for specific policy aims or instruments (Table 3).

**Table 3.** Details of the indicators used to analyze the effectiveness of energy policy initiatives (adapted from [43]).

| Policy Outcomes | Policy Initiatives | Indicators (Unit) |
|---|---|---|
| Decarbonizing energy | Creating solar energy measurement stations; assessing the viability of wind energy systems; increasing the deployment of solar photovoltaic, concentrated solar power, wind, hydro, and nuclear energy; allowing the private sector to purchase and invest in renewable energy; investing in geothermal and waste-to-energy projects; ensuring renewable energy's competitiveness through gradual market liberalization; utilizing atomic energy for saltwater desalination; funding research and development initiatives to support innovation in renewable energy. Details of these renewable energy initiatives are presented in Section 2.2. Table 2 lists the locations of various renewable energy initiatives. | Emissions Intensity: Greenhouse gas emissions per unit of energy consumption ($gCO_2$/million joules). |
| Improving energy efficiency | Modernizing and upgrading the entire power sector; implementing smart grid technology; introducing energy efficiency rating criteria for split air conditioners, lighting, and other household appliances; enforcing improved energy efficiency standards for vehicles; introducing vehicle retirement schemes; charging the iron, cement, and petrochemical industries for inefficient energy use. Section 2.1 of this study discusses various energy efficiency improvement initiatives in detail. | Energy Intensity: Energy consumption per GDP (million joules per USD). |
| Managing energy demand | Developing urban planning guidelines to reduce travel demand; enabling digital transformational programs; increasing the use of insulation materials for new building design and construction; arranging public awareness and education campaigns to reduce energy and water wastage; requiring regular maintenance and repair of buildings (fixing water leaks, etc.) and electrical appliances. These energy demand management initiatives are discussed in Section 2.1. | Energy consumption per capita (gigajoules). |

### 3.2. Temporal Analysis

For trend analysis of each indicator, this study adopts the double exponential smoothing technique because it considers both the mean and the trend component, unlike the single exponential smoothing technique. Moreover, this is a widely used technique for temporal analysis, and many researchers have used it [45,46].

The forecasting equation for the "*m*" period ($F_{n+m}$) under the double exponential smoothing technique is:

$$F_{n+m} = P_n + mb_n \tag{1}$$

here, $P_n$ is the intercept (forecasted); $b_n$ is the slope (forecasted). $P_n$ and $b_n$ can be obtained from the following equations:

$$P_n = \alpha y_n + (1 - \alpha)(P_{n-1} + b_{n-1}) \text{ for } 0 \le \alpha \le 1 \tag{2}$$

$$b_n = \lambda(P_n - P_{n-1}) + (1 - \lambda) b_{n-1} \text{ for } 0 \le \lambda \le 1 \tag{3}$$

here, $\alpha$ and $\lambda$ represent the data smoothing factor and trend smoothing factor, respectively.

### 3.3. Econometric Analysis

This study carried out an econometric analysis to understand the short- and long-term relationships among various drivers of energy sector emissions, including gross domestic product (GDP), total energy consumption (TEC), population (Pop), and net foreign direct investments inflows (FDI). The *EViews* statistical package was used for developing the vector error correction model (VECM) and performing the short- and long-term Granger causality tests; this was chosen because it is a tool that is widely used by researchers for econometric analysis [34]. The details of the development of the VECM are outlined in the following subsections.

#### 3.3.1. Model Specification

In this study, energy sector emissions (GHG) comprise the dependent variable, and GDP, TEC, Pop, and FDI are the independent variables. While choosing independent variables for energy sector emissions, the availability of time series data for each independent variable is ensured because it is a requirement for developing VECM.

The model for this study is specified below:

$$GHG_t = \delta + \beta_1 \, GDP_t + \beta_2 \, TEC_t + \beta_3 \, Pop_t + \beta_4 \, FDI_t + \varepsilon_t \tag{4}$$

here, $\delta$ denotes intercept; $t$ denotes year; $\varepsilon_t$ denotes constant error term; $\beta_1$–$\beta_4$ denote coefficients for GDP, TEC, Pop, and FDI, respectively. Given that it is important to show that all the explanatory variables used in this study collectively describe the dependent variable, this study examines the R-square value for the proposed model. Information on the R-square value is presented in Table 4.

Given that the stationarity test of time series data for dependent and independent variables is essential for VECM, the logarithmic form of Equation (4) is presented in Equation (5).

$$Ln(GHG_t) = Ln(\delta) + \beta_1 \, Ln(GDP_t) + \beta_2 \, Ln(TEC_t) + \beta_3 \, Ln(Pop_t) + \beta_4 \, Ln(FDI_t) + \varepsilon_t \tag{5}$$

#### 3.3.2. VECM Development

There are three steps in the development of a VECM. Firstly, the presence of unit roots in various time series data needs to be examined. Secondly, the existence of cointegration among variables must be determined. Finally, a VECM is developed, and the Granger causality test is performed to examine variables' short- and long-term relationships.

For the unit root test, this study performed the ADF (augmented Dickey–Fuller) and the PP (Phillips–Perron) tests because there is a notion that, despite being widely used, the ADF test often has limited capability to reject a unit root [34]. In addition,

research claimed that combining the ADF test with the PP test provides more robust results [47]. Developing a VECM requires at least two variables among the dependent and independent variables that are co-integrated among themselves. Similarly to other researchers, this study used Johansen's cointegration test for examining the cointegration among variables [34]. This study examined cointegration among variables based on trace statistics and maximum eigenvalue statistics. A VECM was then developed based on the unit root test and cointegration test results. Long- and short-term causal relationships were investigated using the Granger causality test.

### 3.4. Details of the Variables: Data Definition, Source, Units, and Period

Cross-sectional time series data accounting for 1981–2019 were used for this study. A total of five variables were considered: GHG, GDP, TEC, Pop, and FDI. Data were obtained from various sources, and the details of these data sources, along with their units, are presented in Table 4.

**Table 4.** Details of the time series data used for the study (adapted from [47]).

| Data | Description | Unit | Period | Source |
|---|---|---|---|---|
| **Energy Sector Emissions (GHG)** | Total amount of carbon dioxide emissions from the energy sector. | Million tonnes of carbon dioxide | 1981–2019 | Historical greenhouse gas emissions: Saudi Arabia [48]. |
| **Gross Domestic Product (GDP)** | Monetary value of a country's total domestic product in a year. | USD | 1981–2019 | World Development Indicators: Databank [49]. |
| **Total Energy Consumption (TEC)** | Energy consumed in a year. | Quad joules | 1981–2019 | International Energy Agency: Data explorer [50]. |
| **Population (Pop)** | Total number of people in Saudi Arabia. | Person | 1981–2019 | World Development Indicators: Databank [49]. |
| **Foreign Direct Investment (FDI)** | Net inflows of Foreign Direct Investment in Saudi Arabia in a year. | USD | 1981–2019 | World Development Indicators: Databank [49]. |

## 4. Results and Discussion

This section shows how far The Kingdom's energy policies have progressed between 1981 and 2019 and forecasts future trends. This section also presents the causal relationship among various drivers of energy sector emissions.

### 4.1. Effectiveness of Energy Policies

The effectiveness of various energy policies is presented below under three policy outcomes: decarbonizing energy, improving energy efficiency, and managing energy demand.

#### 4.1.1. Decarbonizing Energy

The Kingdom has introduced several decarbonization initiatives across the country. Increased investment has been made in various low-carbon and renewable energy infrastructures, including solar photovoltaic, concentrated solar power, wind power, hydropower, and nuclear energy. Attempts have been made to free the energy market to compete in the renewable energy market. Public and private partnerships have been encouraged to support competition. Funds have been allocated to various research and development initiatives.

Results: Decarbonizing Energy

All the initiatives outlined in the previous section seem to have contributed to a slight decrease in emissions intensity. Emissions intensity has seen a moderate decreasing trend despite an increasing trend in energy sector emissions. For example, in 1981, emissions per unit of energy were around 113 $gCO_2$/million joules, but they decreased to 54 $gCO_2$/million joules in 2019 (Figure 3). Findings reveal that, if such a trend continues, emissions intensity in 2030 will decrease to 47 $gCO_2$/million joules. On the other hand, between 1981 and 2019, emissions from the energy sector increased from 241 million tons of $CO_2$ ($MtCO_2$)

to 582 MtCO$_2$—an increase of approximately 137%. Forecasting shows that energy sector emissions will reach 735 MtCO$_2$ under the current trend in 2030.

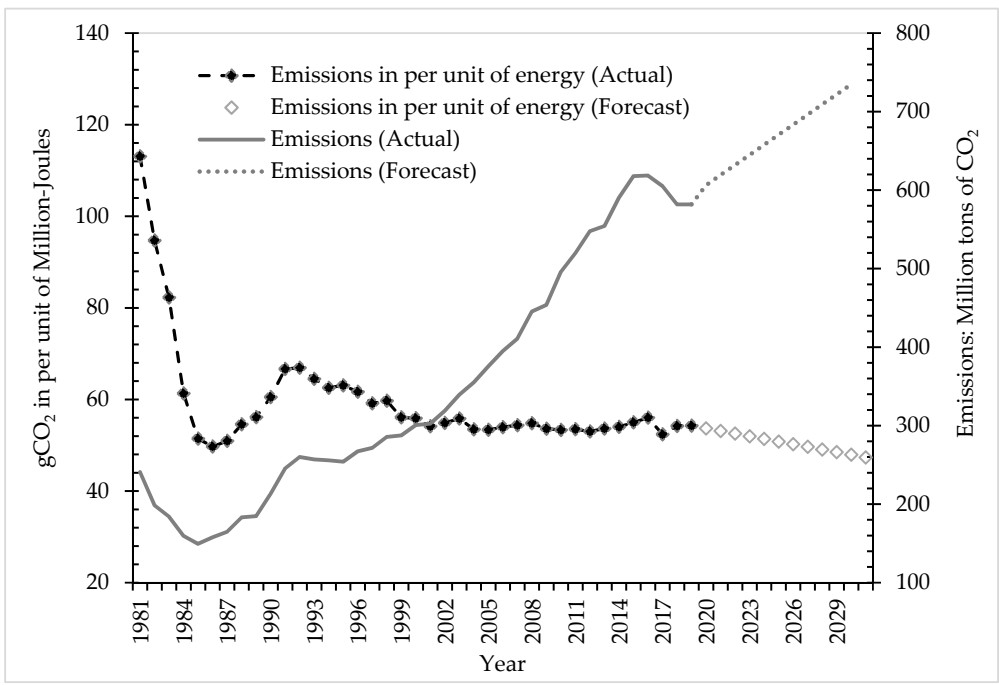

**Figure 3.** The trend of emissions intensity and energy sector emissions.

Commentary: Decarbonizing Energy

A slightly decreasing trend in emissions intensity despite a rapid rise in energy sector emissions indicates that The Kingdom has been progressing steadily towards energy decarbonization. However, work must be carried out to further reduce emissions intensity at a rapid pace. With the current decrease in emissions intensity and trends of renewable energy integration by 2030, achieving the energy decarbonization targets will be difficult.

4.1.2. Improving Energy Efficiency

The Kingdom has given priority to improving energy efficiency across various sectors as the country modernizes the power sector infrastructure and implements smart grid technology to bring efficiency to the power sector. Energy efficiency ratings have been introduced for all building materials and electric appliances, including air conditioners, lighting, heaters, thermal insulators, washing machines, refrigerators, and freezers. The CAFE guidelines have been introduced for light vehicles to improve energy efficiency in the transport sector. In addition, large-scale investments have been made to build public transport infrastructure across the country. Energy efficiency standards are also established for the iron, cement, and petrochemical industries, and fees/charges are introduced for inefficient energy consumption rates.

Results: Improving Energy Efficiency

The effect of the initiatives listed above is quite visible in Figure 4. Even though there have been many fluctuations, mostly due to changes in GDP outputs in some years, the overall energy intensity has decreased. For example, in 1988, the country's energy intensity was the highest, i.e., around 38 million joules/GDP, but it dropped to 14 million joules/GDP by 2019 and will likely decrease to 9 million joules/GDP if the current trend continues. In contrast, the country's GDP rapidly increased from USD 166 billion in 1981 to USD 793 billion in 2019. If the current trend continues, the country's GDP will rise to USD 1008 billion by 2030.

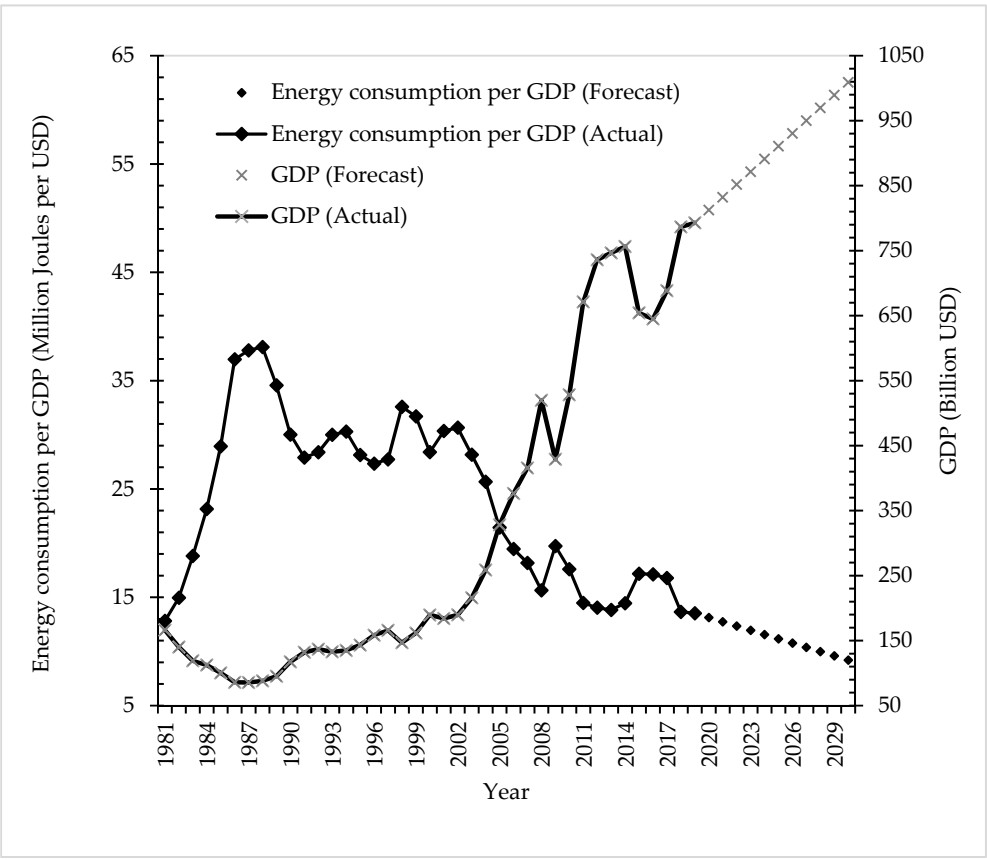

**Figure 4.** The trend of energy intensity and GDP.

Commentary: Improving Energy Efficiency

An overall decrease in energy intensity despite a rapid rise in GDP indicates that The Kingdom's GDP is growing in non-energy-intensive sectors. As a result, the country is showing good progress towards its energy efficiency improvement targets. However, it is essential to note that energy intensity in recent years has been fluctuating, and the rate of decrease in energy intensity over the last few years is relatively low. Therefore, more interventions are required to target other sectors, and lessons learned from various research and innovation experiences need to be implemented on a large scale.

### 4.1.3. Energy Demand Management

Energy demand management is often one of the most cost-effective approaches for reducing energy consumption and energy sector emissions. The Kingdom has taken various initiatives to reduce the energy demand of multiple sectors. It has developed urban planning guidelines to reduce travel demand and introduced various digital transformational programs to reduce travel-related energy consumption. Public education campaigns have been launched to build awareness among the population to reduce energy and water wastage. Policy initiatives ensure that people regularly maintain and repair their buildings (fixing water leaks) and electrical appliances.

Since proper insulation can help reduce energy consumption, policy regulations drive people to use insulating materials in the design and construction of new buildings. Moreover, The Kingdom reformed its energy price in 2018 to halt wasteful energy consumption and reduce the increased energy consumption growth. Research shows that energy price reforms helped the country reduce residential and gasoline energy consumption. As a result, the savings in electricity and gasoline were around SAR 3.8 billion (USD 1 billion) and SAR 8.8 billion (USD 2.34 billion) yearly, respectively [51].

Results: Energy Demand Management

The initiatives taken by the government seem to be fairly ineffective in managing energy demand as the per capita energy consumption has increased from 207 gigajoules in 1981 to 313 gigajoules in 2019 (Figure 5). Findings show that there has been an almost static or slightly increasing trend in per capita energy consumption; under the current trend, it will reach 320 gigajoules by 2030. Increasing total population growth in The Kingdom is a major challenge for the country's energy demand management. Between 1981 and 2019, the total population of Saudi Arabia increased from 10 million to 34 million; with the current trend, the number will reach 41 million by 2030.

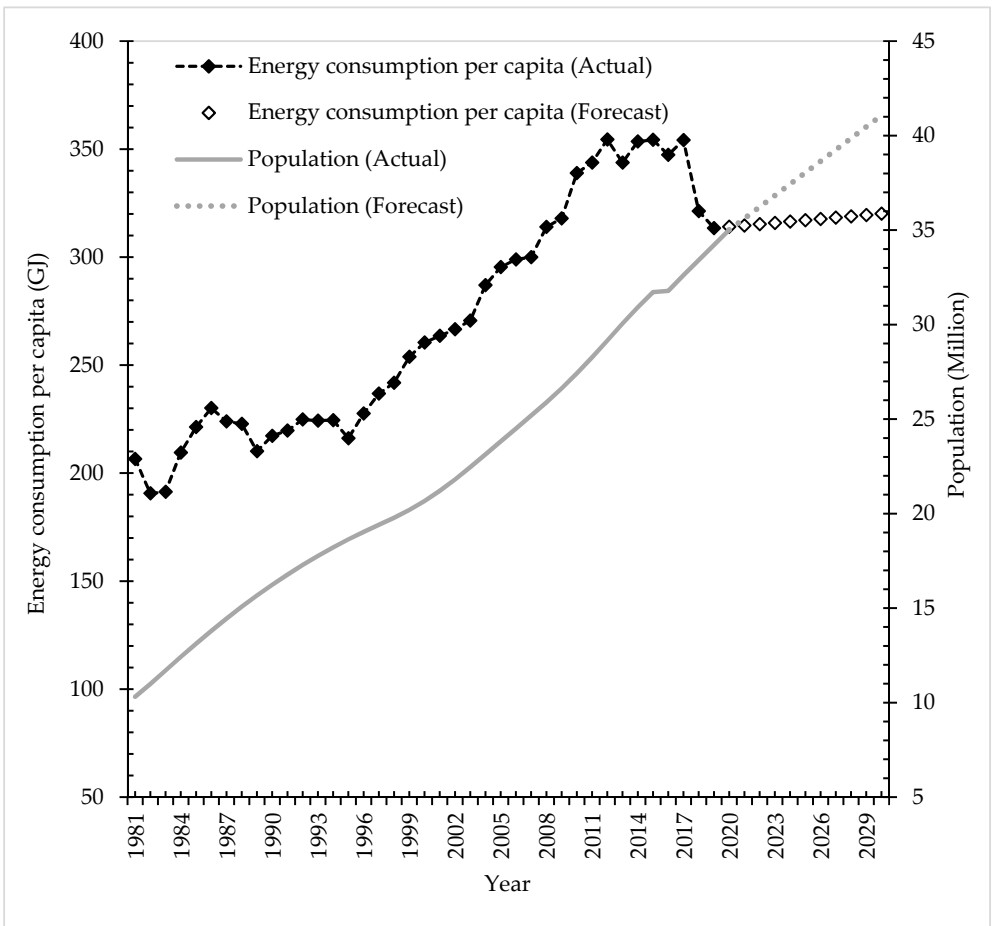

**Figure 5.** The trend of per capita energy consumption and population growth.

Commentary: Energy Demand Management

It is expected that, due to the increasing growth in population, the progress toward reducing energy consumption has become insignificant. This partly indicates that measures taken for managing energy demand are not strong enough to change people's energy-consumption behavior. Most policy initiatives seem voluntary and do not push people to choose prudent energy-consumption behavior. To bring change and achieve the targets under the Saudi Vision 2030, the government needs to rethink its policy measures related to energy demand management and come up with practical solutions; these could be inspired by those which are successfully being used, or which have been proposed for use, in other countries.

*4.2. Econometric Analysis*

This section is divided into two subsections: one for results, and one for commentary.

4.2.1. Results: Econometric Analysis

Results of the various parts of the econometric analysis, including unit root test, cointegration test, equilibrium relationship, and Granger causality test, are discussed here.

Unit Root (UR) Test

This study performed both the ADF and the PP tests to analyze the presence of unit roots in the data. In other words, the ADF and the PP tests were used to understand each variable's time series data stationarity. The results presented in Table 5 show that each variable was found to be nonstationary "At level" and stationary "At first difference", under either the ADF test or the PP test. This means that each variable qualified for use in the development of the VECM.

**Table 5.** Results: Unit root test.

| At Level | Test Statistics (ADF) | | Test Statistics (PP) | |
|---|---|---|---|---|
| | **Intercept** | **Intercept and Trend** | **Intercept** | **Intercept and Trend** |
| GHG | −0.33 (2) | −2.30 (2) | 0.40 (3) | −3.46 * (3) |
| GDP | 0.89 (0) | −2.00 (0) | 0.76 (2) | −2.01 (2) |
| TEC | −1.78 (3) | −4.97 *** (7) | 0.03 (3) | −1.78 (3) |
| Pop | 1.99 (2) | −4.87 *** (8) | 1.23 (4) | −0.53 (4) |
| FDI | −2.17 (2) | −2.56 (2) | −1.84 (2) | −1.96 (2) |
| **At first difference** | | | | |
| GHG | −4.17 *** (2) | −3.91 ** (2) | −4.15 *** (5) | −3.86 ** (4) |
| GDP | −4.99 *** (0) | −5.29 *** (2) | −4.97 *** (2) | −5.24 *** (2) |
| TEC | −1.46 (1) | 0.13 (2) | −5.78 *** (3) | −5.69 *** (3) |
| Pop | −3.35 ** (2) | −3.67 ** (0) | −3.30 ** (2) | −3.61 ** (2) |
| FDI | −4.26 *** (2) | −4.19 ** (2) | −4.26 *** (0) | −4.19 ** (0) |

Significance levels at 0.1, 0.05, and 0.01 are symbolized by *, **, and ***, respectively. Lag lengths were identified using Schwarz info criterion (SIC) and are presented within parentheses.

Cointegration Test

For developing a VECM, it is a prerequisite that at least two variables are integrated. This means there should be at least one cointegrating equation in the model. Based on the trace statistics and maximum eigenvalue statistics of Johansen's cointegration test (Table 6), this study confirmed at least one cointegrating equation at the 0.01 level. This validated the VECM developed in this study.

**Table 6.** Results: Johansen's test of cointegration.

| No. of Cointegration Equation | Hypothesis | Trace Statistics | Maximum Eigenvalue Statistics |
|---|---|---|---|
| $r = 0$ | No cointegration equations | 105.29 *** | 46.41 *** |
| $r = 1$ | At most 1 cointegration equation | 58.89 *** | 27.58 *** |
| $r = 2$ | At most 2 cointegration equations | 29.80 ** | 21.13 * |
| $r = 3$ | At most 3 cointegration equations | 13.8 * | 13.8 * |
| $r = 4$ | At most 4 cointegration equations | 0.001 | 0.001 |

Note: ***, **, and * symbolize significance levels at 0.01, 0.05, and 0.1, respectively.

Equilibrium Relationship

Table 7 presents the coefficients of each variable along with the signs. GHG was used as the dependent variable, and GDP, TEC, Pop, and FDI were used as the independent variables. A positive sign for each variable indicates a positive relationship between each independent variable and the dependent variable. This means GHG was found to increase with increases in GDP, TEC, POP, and FDI for Saudi Arabia. An R-squared value of 0.80 implies that independent variables adequately explain the variations of the dependent variable.

**Table 7.** Cointegrating equations.

| Dependent Variable: GHG | | | |
|---|---|---|---|
| **R-Squared Value: 0.80** | | | |
| **Explanatory Variable** | **Coefficient** | **Standard Error** | **T Statistics** |
| Constants | 333.48 | 497.73 | 0.67 |
| GDP | $6.18 \times -10$ | $1.1 \times -10$ | 5.44 |
| TEC | 0.13 | 0.02 | 6.71 |
| POP | 60.22 | 6.75 | 8.92 |
| FDI | $3.96 \times -10$ | $6.6 \times -10$ | 0.60 |

Granger Causality Test: Short- and Long-Term Relationships

Using the Granger causality test, this study explored the causal relationships between dependent and independent variables in the short and long term. In addition to the causal relationships, Granger causality helps identify any relationship's direction. For example, Table 8 shows that the total energy consumption and FDI inflows of Saudi Arabia have a causal long-term relationship with energy sector emissions, and these relationships are significant at a 0.1 level. Regarding short-term causality, this study reveals that there is a unidirectional causal relationship running from total energy consumption to energy sector emissions. From Table 8, it is also evident that the short-term relationship between population and energy sector emissions is bi-directional. This means population growth causes an increase in energy sector emissions, while increased emissions also lead to population growth. Findings also show that FDI causes GDP, and the relationship is unidirectional in the short term. Therefore, there is also a unidirectional causal relationship running from population and FDI to total energy consumption in the short term. In addition, Table 8 indicates that increases in GDP and FDI contribute to population growth, but the relationship is unidirectional, running from GDP and FDI to population.

**Table 8.** Causality test results (Short- and long-term relationships).

| | | **Granger Causality—Short-Term: F Statistics** | | | | **Granger Causality—Long-Term: T Statistics** |
|---|---|---|---|---|---|---|
| | **Ln (GHG)** | **Ln (GDP)** | **Ln (TEC)** | **Ln (POP)** | **Ln (FDI)** | **Error Correction Terms** |
| Ln (GHG) | | 1.68 (0.21) | 2.31 * (0.1) | 3.98 ** (0.03) | 2.15 (0.14) | −4.00 *** (0.001) |
| Ln (GDP) | 0.58 (0.57) | - | 0.74 (0.49) | 1.04 (0.37) | 3.33 ** (0.05) | 1.50 (0.15) |
| Ln (TEC) | 0.48 (0.62) | 0.23 (0.80) | - | 11.69 *** (0.0003) | 6.95 *** (0.004) | 1.96 * (0.06) |
| Ln (POP) | 3.13 * (0.06) | 14.49 *** (0.0001) | 1.27 (0.30) | - | 2.52 * (0.1) | 1.52 (0.14) |
| Ln (FDI) | 0.23 (0.80) | 0.44 (0.65) | 0.65 (0.53) | 0.13 (0.88) | - | 1.72 * (0.09) |

Note: ***, **, and * symbolize significance levels at 0.01, 0.05, and 0.1, respectively. Probability values or p-values are placed within the parentheses.

### 4.2.2. Commentary: Econometric Analysis

From the econometric analysis, it is evident that some explanatory variables are co-integrated among themselves, and energy consumption and FDI inflows could be key contributors to energy sector emissions in the long term. To decouple energy consumption from emissions, focus needs to be placed on increased renewable energy use, energy demand management, and energy efficiency. Although the government is actively considering heavy investments in each of these, these initiatives must be implemented without further delay.

In the short term, the government should ensure that FDI inflows are contributing towards low-carbon GDP growth. Given that population growth is also contributing to short-term energy sector emissions, the government could focus on rapid urbanization with high-density urban development. This is also likely to reduce energy consumption.

### 5. Policy Implications

The energy sector of The Kingdom of Saudi Arabia remains dependent on fossil fuels. The current share of renewable and nuclear energy is insignificant in the national energy portfolio. GDP, FDI, and population are strongly connected, and these drive energy sector emissions. Although energy consumption per GDP has been decreasing, the growth in GDP diminishes significant reduction in total emissions. The Kingdom's energy efficiency initiatives, including relevant standards, guidelines, and educational program development, have contributed to reducing emissions. In addition, the reform in electricity generation through adopting a combined cycle has been causing a reduction in emissions. The long-term trend of per capita energy consumption is positive, but the value is slightly decreasing in recent years. The energy sector emissions have experienced a significant reduction in recent years, mainly due to the reform in energy prices.

Based on the above discussion, it is imperative for The Kingdom to adopt a multifaceted approach to ensure significant emission reduction in the energy sector and to reach the targets for net-zero carbon emissions. The reduction in emissions through energy efficiency and demand management will reach its limits due to technological, investment, and social welfare considerations. This reduction will support reaching net-zero carbon emissions through renewable and nuclear energy generation. However, due to the intermittency and the non-homogenous spatial distribution of renewable energy resources, and the availability of fossil fuels, the complete phase-out of fossil fuels is not expected for The Kingdom. Instead, the development of CCS, CCUS, afforestation, and carbon-trading systems will possibly benefit future emission scenarios. However, the successes of CCS and CCUS remain limited for the foreseeable future. The afforestation initiative will aid in carbon removal and offset carbon emissions. However, this program will have hydrological and ecological issues along with a significant water-use footprint [52]. The major corporations of The Kingdom—including Saudi Aramco, Saudia airlines, ACWA power, Ma'aden, and ENOWA—have been expressing their interest in voluntary carbon markets. Therefore, the roadmap to net-zero carbon emissions by 2060 is expected to be a dynamic process that will evolve with the success of new technologies, significant changes in the national energy portfolio, carbon sink development, and offset schemes through carbon trading.

### 6. Conclusions

This paper provides a holistic overview of the various energy efficiency and renewable energy initiatives undertaken by the government of Saudi Arabia to reduce GHG emissions in the energy sector. The efficacy of energy-related policies and initiatives is evaluated using an indicator-based approach. Then, this study performed temporal and econometrics analyses to understand the trend and the causal relationship among various drivers of energy sector emissions. It is evident from the analyses that the government has made some progress toward achieving its energy efficiency targets. Still, the progress toward decarbonizing energy and managing energy demand is inadequate, indicating that more emphasis is needed on The Kingdom's renewable energy and energy demand management

initiatives. The renewable energy initiatives seem to cover almost all sorts of renewable and atomic sources, including solar, wind, hydro, geothermal, and waste-to-energy technologies. Despite these initiatives, the progress towards energy decarbonization is insufficient for a few reasons. Firstly, most of the initiatives were undertaken a few years ago, and it might take some time to build infrastructures and generate energy from those renewable sources. Secondly, the scale of energy generation from these renewable initiatives is not significant considering the ambition of the government's plan. This indicates that The Kingdom needs to scale up its renewable energy generation. Finally, many renewable energy initiatives are still under research investigation, and these initiatives will not be implemented for a few years. Therefore, transferring the research outputs into the real world as soon as possible is important if we are to see rapid progress towards energy decarbonization.

The energy demand management initiatives undertaken by the government are not comprehensive enough to ensure a change in people's energy-consumption behaviors. Attention needs to be given to all facilities, assets, activities, and operations for each source of energy consumption, including industries, transportation, buildings, waste, water, and agriculture. Apart from these initiatives, there needs to be a comprehensive monitoring and evaluation framework to easily identify issues and provide tailor-made solutions. For example, a carbon tax or an emissions-trading scheme could be valuable tools for changing people's energy-consumption behavior. Lastly, voluntary approaches are often ineffective in managing energy demand. Therefore, it is crucial to combine voluntary and mandatory techniques to change people's behavior and thus reduce energy consumption.

## 7. Limitations and Future Research

One of the limitations of this research is that some factors of energy sector emissions—such as fossil fuel and renewable energy costs, urbanization rates, technological innovation, and behavioral changes—could not be included in the econometric analysis. This is because the datasets accounting for energy costs and urbanization were stationary at the required level, which is a precondition for developing a VECM. Furthermore, we could not use technological innovation of behavior change in the analyses because a VECM requires time series data, and we could not find such data during the investigation. In addition, this study does not compare its findings with other Gulf Cooperation Council countries such as Bahrain, Kuwait, Oman, Qatar, or the United Arab Emirates. A comparative study with other countries would help us to better understand The Kingdom's progress towards reducing energy sector emissions.

**Author Contributions:** Conceptualization, M.A.H., S.M.R., M.S. and M.M.R.; data curation, M.A.H. and M.S.; formal analysis, M.A.H., M.S. and M.M.R.; funding acquisition, M.M.R.; investigation, M.S., M.S.R., M.A., M.K.I. and M.M.I.; methodology, M.A.H., M.S. and S.M.R.; project administration, M.S. and M.M.R.; resources, M.S., M.A. and M.S.R.; supervision, S.M.R. and M.M.R.; visualization, M.A.H. and M.K.I.; writing—original draft preparation, M.A.H., M.S. and M.M.R.; writing—review and editing, S.M.R., M.S.R. and M.M.I. All authors have read and agreed to the published version of the manuscript.

**Funding:** This research was funded by the Deanship of Scientific Research at King Faisal University (KFU), Al-Ahsa 31982, Saudi Arabia, through Project No. GRANT1442.

**Institutional Review Board Statement:** Not applicable.

**Informed Consent Statement:** Not applicable.

**Data Availability Statement:** The data that support the findings of this study are available from the corresponding author, M.M.R. (mrahman@kfu.edu.sa), upon reasonable request.

**Acknowledgments:** The authors acknowledge the support received from the Deanship of Scientific Research at King Faisal University (KFU), King Fahd University of Petroleum & Minerals (KFUPM), and Al-Imam Mohammad Ibn Saud Islamic University, Saudi Arabia.

**Conflicts of Interest:** The authors declare no conflict of interest.

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
