# Peer review of "A Critical, Temporal Analysis of Saudi Arabia’s Initiatives for Greenhouse Gas Emissions Reduction in the Energy Sector"

_sustainability, doi:10.3390/su141912651_

Round 1

Reviewer 1 Report (Previous Reviewer 1)

Thank you for the opportunity to review this manuscript. This article examined the effectiveness of Saudi Arabia's leadership's numerous energy efficiency and renewable energy projects. I have no reservations regarding the topic or the contents, and it is nearly ready for publishing. However, I recommend some modest revisions before proceeding to the following steps of determining acceptance. 

1) Section 2 contains far too much information about energy efficiency; this section should be explicitly cut. 

2) In sections 2 and 3, all detail information is given as an indicator. Tables 2 and 3 as well as the text should be redirected. 

3) Section 4 should be divided into two parts: results and commentary. Please avoid combining results and commentary in order to address your findings properly. 

4) Because this is an academic study, there is no need to include 'policy implications.' Please move this section into the discussion section. 

5) Please combine sections 5 and 6. 

Section 5 is divided into two sections. Please be cautious when modifying your final manuscript version before re-submission.

Author Response

Reviewer 2 Report (Previous Reviewer 3)

This manuscript needs to be improved significantly. Consider the following remarks carefully:

1) Highlight the novelty of your study. Clearly discuss what the previous studies that you are referring to are. What are the research gaps/contributions?

2) More analysis and interpretation are needed for the short-term, medium-term and long- term goals. There are many indefinite articles missing related to emission reduction methods, and kindly fill the gap with Energy Conversion and Management 2022; 264, 115758.

3) What is the practical and policy implications of the study? Kindly include that as a section before conclusion.

4) Please make sure your conclusions' section underscores the scientific value-added of your paper, and/or the applicability of your results.

Author Response

Reviewer 3 Report (New Reviewer)

The authors studied the KSA initiatives toward GHG emissions reduction. The topic is impressive and worth investigating. However, the following are observed:

- The authors claimed that the GHG emissions of Saudi Arabia are more than three times the global average emission in 2019, but I have not seen any scholarly proof of this claim. Several statements need citations to validate them.

-Another statement said, " it is unclear how effective these initiatives are in improving energy 110 efficiency and enhancing the share of renewable energy for lowering the GHG emissions." However, I know some research work that highlights these areas, some of which are done by these authors but not reported in the current study-they include: https://doi.org/10.1016/j.rser.2011.12.003https://doi.org/10.3390/su14127388,https://doi.org/10.1016/j.jksus.2021.101477, https://doi.org/10.1007/978-981-15-6058-3_5 , etc.

-Under section 3.2, it has been said that many researchers have used double exponential smoothing for Temporal analysis, but only one reference is mentioned [44]. I expect more than a single reference here to qualify the term "MANY."

- Equation 4 under model specification must be justified. The choice of response and explanatory variables.

- The article needs thorough English editing.

-Policy implications and conclusions sections are well written, and limitations and future research are promising.

Congratulations!!

Round 2

Reviewer 2 Report (Previous Reviewer 3)

I am satisfied with the manuscript in its current form and I recommend it for publication.

This manuscript is a resubmission of an earlier submission. The following is a list of the peer review reports and author responses from that submission.

Round 1

Reviewer 1 Report

This manuscript has a tremendous and time-mannerly topic of energy efficiency.  The Authors should consider making a major revision to improve the article. Please include the following tasks in the revision.

1) Strengthen the methodology.

2) Follow the journal format template.

3) Avoid logical saltus with the numerical data and energy policy changes.

4) Use USD currency.

5) Hire an English native speaker editor for proofreading before the next submission.

This manuscript should be entirely revised from the research design step.

Reviewer 2 Report

The abstract is not well written. The contribution of this paper is not clearly stated. What is the objective of the paper, main methods, main results? What are the policy implications of the results.

The empirical analysis includes only few graphs. I expected to see some results based on econometric models to support the novelty of the paper. In this context, the degree of novelty for this research is not enough to publish in this journal.

I do not see a deep link between the concept of sustaianability and the topic of the paper.

The section for methodology description is absent.

Details on how the trend was computed are not provided.

Limits of the researh are not indicated.

Future directions of research are not presented.
